# The Impact of Human Liver Transplantation on the Concentration of Fibroblast Growth Factors: FGF19 and FGF21

**DOI:** 10.3390/ijms26031299

**Published:** 2025-02-03

**Authors:** Marta Budkowska, Ewa Ostrycharz-Jasek, Elżbieta Cecerska-Heryć, Katarzyna Dołęgowska, Aldona Siennicka, Łukasz Nazarewski, Paweł Rykowski, Barbara Dołęgowska

**Affiliations:** 1Department of Medical Analytics, Pomeranian Medical University, Al. Powstańców Wielkopolskich 72, 70-111 Szczecin, Poland; aldona.siennicka@pum.edu.pl; 2Institute of Biology, University of Szczecin, 71-412 Szczecin, Poland; ewa.ostrycharz@phd.usz.edu.pl; 3Doctoral School, University of Szczecin, 70-383 Szczecin, Poland; 4Molecular Biology and Biotechnology Center, University of Szczecin, 71-412 Szczecin, Poland; 5Department of Laboratory Medicine, Pomeranian Medical University, Al. Powstańców Wielkopolskich 72, 70-111 Szczecin, Poland; elzbieta.cecerska.heryc@pum.edu.pl (E.C.-H.); barbara.dolegowska@pum.edu.pl (B.D.); 6Department of Immunology Diagnostics, Pomeranian Medical University, Al. Powstanców Wielkopolskich 72, 70-111 Szczecin, Poland; katarzyna.dolegowska@pum.edu.pl; 7Department of General, Transplant and Liver Surgery, Medical University of Warsaw, ul Banacha 1a, 02-097 Warsaw, Poland; lukasznazarewski1985@gmail.com (Ł.N.); pawel.ryk@wum.edu.pl (P.R.)

**Keywords:** fibroblast growth factors, liver transplantation, ischemia–reperfusion injury

## Abstract

The multitude of processes in which the liver participates makes it vulnerable to many serious diseases, which can lead to chronic organ failure. Modern medicine bases the treatment of end-stage liver failure on liver transplantation. To ensure the proper functioning of the transplanted liver, a balance of cellular and immunological processes and appropriate concentrations of many different factors are necessary, including, among others, fibroblast growth factors (FGFs). Over the last several years, studies have focused on some FGF growth factors, i.e., FGF19 and FGF21. These two growth factors belong to the FGF19 subfamily, and we concentrate on these two factors in our work. These factors diffuse away from the site of secretion into the blood, acting as hormones. FGF19 is a growth factor with a high therapeutic potential, involved in the homeostasis of bile acids necessary to maintain the proper function of the transplanted liver. FGF21, in turn, plays an important role in regulating lipid and glucose homeostasis. This study aimed to evaluate changes in the concentration of growth factors FGF19 and FGF21 in the plasma of 84 patients before, 24 h, and 2 weeks after liver transplantation (ELISA test was used). Additionally, the correlations of the basic laboratory parameters—alanine aminotransferase (ALT), aspartate aminotransferase (AST), gamma-glutamyl transpeptidase (GGTP), alkaline phosphatase (ALP), total bilirubin, C-reactive protein (CRP), albumin and hemoglobin (Hb)—with FGF19 and FGF21 were determined. Our studies noted statistically significant changes in FGF19 and FGF21 concentrations before, 24 h, and 2 weeks after liver transplantation. The highest values for FGF19 before liver transplantation and the lowest values 24 h after this surgery were observed for FGF21; the highest concentrations were observed the day after liver transplantation, and the lowest were observed immediately before surgery. Observations of increases and decreases in the concentration of the examined factors at individual time points (before and after transplantation) allow us to suspect that FGF19 has an adaptive and protective function toward the transplanted liver. At the same time, FGF21 may affect the regenerative mechanisms of the damaged organ.

## 1. Introduction

The liver is a critical center of many physiological processes. It is exposed to damage that can be a direct or indirect cause of many serious diseases such as hepatocellular carcinoma (HCC), cirrhosis caused by hepatitis B (HBV) or C (HCV), alcoholic liver disease (ALD), nonalcoholic steatohepatitis (NASH), nonalcoholic fatty liver disease (NAFLD), primary sclerosing cholangitis (PSC), primary biliary cholangitis (PBC), autoimmune hepatitis (AIH), and others [1]. Over time, these diseases can lead to the end-stage failure of this organ and, consequently, to life-threatening conditions. In such cases, transplantation might be the only chance for the patient [2,3]. The liver is one of the most frequently transplanted organs, and these operations are characterized by high survival rates and an improvement in the quality of life of patients [2,4]. Due to the growing demand for this organ and, at the same time, its constant shortage, it seems very important to learn the mechanisms that support its normal functions immediately after transplantation. This seems particularly important in the context of damage to the transplanted organ caused by the mechanism of ischemia–reperfusion injury (IRI). This mechanism is the cause of early organ failure (<10% of all cases) and can lead to acute or chronic graft rejection. Liver damage occurs due to tissue hypoxia, which impairs the metabolism and function of the hepatocyte through the depletion of ATP resources, the accumulation of sodium, and the formation of cellular edema. The IRI mechanism also involves interleukin 1 (IL-1), interleukin 6 (IL-6), interferon-gamma (IFN-γ), platelet-activating factor (PAF), and tumor necrosis factor α (TNF-α) [4,5,6]. In recent years, researchers have discovered growth factors that can positively affect the functions of transplanted organs (especially kidneys and the liver), e.g., fibroblast growth factors, especially the FGF19 subclass [7,8,9,10,11,12,13,14,15]. The FGF19 subclass, including FGF19, FGF21, and FGF23, is a unique subfamily of fibroblast growth factors that shows a weak affinity for heparan sulfate (HS). This allows facilitated diffusion from the site of secretion into the blood, where these factors perform hormonal functions [16,17]. For this reason, the FGF19 subclass is also called the “endocrine group” [18,19,20,21,22,23]. Proteins from the FGF19 subfamily affect the enterohepatic circulation of bile, regulate glucose and lipid metabolism, and maintain the homeostasis of phosphate, vitamin D, and bile acid metabolism [18,20,22,23,24]. Our study aimed to determine changes in the concentration of growth factors FGF19 and FGF21 in patients before and after liver transplantation (24 h, 2 weeks) and to assess whether any changes in the concentration of the tested factors may affect the normal function of the transplanted liver or its damage in IRI mechanism. The obtained results were also compared with the results of the concentrations of these factors in healthy subjects as the control group. In addition, the correlations of basic laboratory parameters with the growth factors (FGF19, FGF21) were determined.

## 2. Results

### 2.1. Analysis of Average FGF19 Concentration in Patients Before and After Liver Transplantation and in the Healthy Control Group—Statistics Data

FGF19 plasma concentrations differed significantly between the period before and after transplantation, as well as compared to the control group. Our studies showed that the highest concentration of FGF19 in the plasma occurred before transplantation (0 h) (1065.9 ± 423 pg/mL). On the first day after the transplantation (24 h), FGF19 concentration in plasma was 3.8-fold lower (*p* < 0.0001 vs. 0 h) compared to baseline values (0 h), reaching the value of 280.4 ± 132 pg/mL. Whereas 2 weeks after the transplantation, the level of FGF19 in the plasma increased to 497.3 ± 191 pg/mL (1.7-fold, *p* < 0.0001 vs. 24 h after the transplantation). The average FGF19 plasma concentration in the control group was 7-fold lower than in patients before liver transplantation, and it was 151.0 ± 47.7 pg/mL. (Figure 1). Statistical analysis confirmed the occurrence of statistically significant differences in FGF19 concentrations before and after liver transplantation, as well as compared to the control group and each time point in transplant patients (*p* < 0.0001).

### 2.2. FGF19 Concentration Before and After Liver Transplantation Depending on Primary Disease

FGF19 concentrations in plasma, depending on the primary disease, differed significantly between the period before and after transplantation in all studied diseases and compared to the control group. (Table 1). Our studies showed that in all diseases, the highest concentration of FGF in the plasma occurred before transplantation (0 h). In contrast, the lowest concentration in all patients occurred 24 h after transplantation. It gradually increased, reaching 2 weeks after transplantation, where the values on average were 2.1-fold lower than the baseline values (before transplantation). However, the lowest FGF19 concentrations were still observed in the control group (Figure 1).

Statistical analyses revealed no significant differences in FGF19 concentrations among disease entities at any of the examined time points (before transplantation, 24 h post-transplantation, and 2 weeks post-transplantation).

### 2.3. Analysis of Average FGF21 Concentration in Patients Before and After Liver Transplantation and in the Healthy Control Group—Statistics Data

FGF21 plasma concentrations differed significantly between the period before and after transplantation, as well as compared to the control group. Our studies showed that, in the group of patients, the lowest concentration of FGF21 in the plasma occurred before transplantation (0 h) (466.9 ± 408 pg/mL). The first day after the transplantation (24 h), FGF21 concentration in plasma reached the highest concentration of 3258.4 ± 4804 pg/mL (*p* < 0.0001 vs. 0 h) and was 7-fold greater compared to baseline values (0 h), whereas 2 weeks after the transplantation, the level of FGF21 in the plasma decreased to 2144.6 ± 2743 pg/mL (0.7-fold, *p* < 0.0001 vs. 24 h after the transplantation). The average FGF21 plasma concentration in the control group was 1.6-fold lower than in patients before liver transplantation, and it was 290.59 ± 47.7 pg/mL (Figure 2). Statistical analysis confirmed the occurrence of statistically significant differences in FGF21 concentrations before and after liver transplantation, as well as compared to the control group and each time point in transplant patients (*p* < 0.0001).

### 2.4. FGF21 Concentration Before and After Liver Transplantation Depending on the Primary Disease

FGF21 concentrations in plasma, depending on the primary disease, differed significantly between the period before and after transplantation in all studied diseases, as well as compared to the control group (Table 2). The lowest FGF21 concentration was observed in the control group and was statistically significant in almost all diseases studied except ALD, PBC, and PSC. Our studies showed that in all diseases, the lowest concentration of FGF21 in the plasma occurred before transplantation (0 h). The highest concentration of FGF21 was observed 2 weeks after transplantation in the group of patients with AIH and ALD. In contrast, in the remaining disease entities (PBC, PSC, HBV, HCV, and HCC), the highest concentration of FGF21 was observed 24 h after transplantation (Figure 2).

In the case of FGF21, the concentration in the plasma of patients, both before and after liver transplantation, varied depending on the disease. Before transplantation, the concentration of FGF21 differed significantly between AIH and HBV (*p* = 0.046), HCV and ALD (*p* = 0.049), HCV and PSC (*p* = 0.027), HCV and HBV (*p* < 0.0001), and HBV and HCC (*p* = 0.0017). The highest serum concentration of FGF21 was observed in HCV patients, and the lowest was observed in the HBV group (Table 2, Figure 3A).

Notably, 24 h after transplantation, the concentration of FGF21 differed significantly between ALD and HCC (*p* = 0.0018), ALD and PBC (*p* = 0.025), ALD and HCV (*p* < 0.0001), and HCV and HBV (*p* = 0.003). The highest serum concentration of FGF21 was observed in HCV patients, and the lowest was observed in the ALD group (Table 2, Figure 3B).

And 2 weeks after transplantation, the concentration of FGF21 differed significantly between AIH and HCC (*p* = 0.036), AIH and HBV (*p* < 0.0001), AIH and PSC (*p* < 0.0001), AIH and PBC (*p* = 0.013), ALD and HBV (*p* = 0.047), ALD and PSC (*p* = 0.015), PSC and HCV (*p* = 0.0012), and HBV and HCV (*p* = 0.004). The highest plasma concentration of FGF21 was observed in AIH patients, and the lowest was observed in the PSC group (Table 2, Figure 3C).

### 2.5. FGF19 and FGF21 Concentration Before and After Liver Transplantation Depending on the Primary Disease, Patient’s Gender, and Age Group

Our research did not show statistically significant differences in the concentrations of FGF19 and FGF21 depending on gender and the period of blood collection (before the transplantation (0 h), as well as 24 h and 2 weeks after its execution), except in three situations. There was a statistically significant difference in FGF19 concentration between men and women for HCC before transplantation (0 h) (892.8 ± 369 pg/mL and 1338 ± 435 pg/mL, respectively; *p* = 0.04) and 24 h after transplantation (241.5 ± 90 pg/mL and 461.3 ± 166 pg/mL, respectively; *p* = 0.004) (Figure 4A). In addition, our study showed differences between male and female groups compared to the control group at both time points for HCC. In the case of HCC (0 h), FGF19 levels were 5.9-fold higher in males than in the control group, while in females, they were 8.9-fold higher compared to the control group; whereas 24 h after transplantation, in the HCC group, the FGF19 level was 1.6-fold higher in men and 3-fold higher in women compared to the control group (Figure 4B). In the case of FGF21, the differences in concentrations between men and women were demonstrated only 24 h after transplantation in the group of people suffering from HCV (*p* = 0.014). Similarly to FGF19, FGF21 levels were significantly higher than in the control group in both men (*p* < 0.001) and women (*p* < 0.001) (Figure 4C).

Depending on the age group (equally or below 45 years and above 45 years), FGF19 and FGF21 concentration differences were observed only in patients with PBC and AIH in the pre-transplant period (0 h). In the group of younger patients, the concentration of FGF19 was 1.9-fold higher than in patients over 45 years of age (1799.3 ± 576 and 910.5 ± 273, respectively; *p* = 0.028). Additionally, FGF19 concentration in younger patients was 11.9-fold higher, whereas in older patients, it was 6.4-fold higher compared to the control group (Figure 5A). In the PBC group, before transplantation (0 h), the concentration of FGF21 was 3-fold higher in patients below or equal to 45 years old (676.0 ± 86 pg/mL and 219.8 ± 45 pg/mL, respectively; *p* < 0.0001) (Figure 5B). Our study also showed higher levels of FGF21 in the younger group compared to the control group (2.3-fold change), while in people over 45, the FGF21 level was 1.3-fold lower than in the control group. In contrast to PBC patients, AIH patients observed a 2.7-fold lower concentration of FGF21 in the equally or below 45 years group in comparison to the over 45 years group (319.5 ± 170 pg/mL and 879.0 ± 387 pg/mL, respectively; *p* = 0.007). However, the difference in the FGF21 level in AIH compared to the control group was demonstrated only in the group of people aged over 45, which was 3-fold higher (Figure 5C).

A statistical analysis of the relationships between FGF19 and FGF21 concentrations, MELD score, and operation time revealed no significant correlations except for two cases. In HCV patients, the FGF19 level measured 24 h after surgery exhibited a negative correlation with the length of surgery time (r = −0.7748, *p* = 0.041) (Figure 6A). In contrast, the FGF21 level measured 2 weeks post-transplantation in the same group of patients showed a positive correlation with the MELD score (r = 0.7818, *p* = 0.008) (Figure 6B).

### 2.6. Correlations of Basic Laboratory Parameters with FGF19 and FGF21

As part of the statistical analysis, Pearson or Spearman’s correlation test, depending on the normal distribution, was performed for laboratory parameters (CRP, Hb, total bilirubin, ALP, ALT, AST, GGTP, and albumin) with the FGF19 and FGF21.

The FGF19 concentration exhibited a positive correlation with CRP (r = 0.7308, *p* = 0.005) before transplantation (0 h) in the AIH group and 2 weeks after transplantation in the PBC group (r = 0.7237, *p* = 0.028) and the HCC group (0.4842, *p* = 0.036). However, a negative correlation between FGF19 and CRP was also observed 24 h after transplantation (r = −0.6485, *p* = 0.043) in patients with HCV. In the case of hemoglobin, we observed a positive correlation between the concentration of Hb and the concentration of FGF19 24 h after transplantation in HCC patients (r = 0.4889, *p* = 0.033) and 2 weeks after transplantation in HBV patients (r = 0.6997, *p* = 0.011). Additionally, 2 weeks after transplantation, we observed a positive correlation between FGF19 and ALT (r = 0.4906, *p* = 0.033) and total bilirubin (r = 0.4781, *p* = 0.038) in the HCC group. In the same period, we observed a negative correlation with the levels of FGF19 and AST (r = −0.5821, *p* = 0.047) in the ALD group.

In the case of FGF21, we did not find any statistically significant correlations in the period before transplantation (0 h). However, 24 h after transplantation, we observed a negative correlation for FGF21 with the levels of ALT (r = −0.7091, *p* = 0.022) and hemoglobin (r = −0.7295, *p* = 0.017) in HCV patients. Additionally, in the same period, PSC patients had a positive correlation between FGF21 and CRP (r = 0.7618, *p* = 0.017). Two weeks after transplantation in ALD patients, we observed a positive correlation between the enzymes. ALP (r = 0.05804, *p* = 0.047) and GGTP (r = 0.7343, *p* = 0.007). In the same period in PSC patients, we observed a positive correlation in the levels of CRP (r = 0.7682, *p* = 0.016) and negative in albumin levels (r = −0.8608, *p* = 0.003).

## 3. Discussion

### 3.1. Changes in FGF19 Concentration in Patients Before and After Liver Transplantation (24 h, 2 Weeks) and Comparison in Those Changes with the Healthy Control Group

Our studies show that the average FGF19 plasma concentration in the control group was several times lower than in patients before liver transplantation, and it was 151.0 ± 47.7 pg/mL. The concentration of FGF19 in plasma collected on EDTA from a group of healthy subjects in the study by Dostalova et al. performed using the same method and from the same manufacturer is, on average, 205.3 ± 25.97 pg/mL [25], which is a slightly higher result than we obtained. However, it should be emphasized that the concentration of FGF19 in healthy people may vary depending on several factors, such as age, gender, time of day, diet, and other physiological conditions [11,18,20,26]. It is generally accepted that in the plasma/serum of healthy people, the concentration of FGF19 ranges from 50 to 200 pg/mL, which makes our result correct and consistent with other reports [11,24,27,28,29].

Our average FGF19 concentration in the control group was much lower than in patients before liver transplantation (151.0 ± 47.7 vs. 1065.9 ± 423). Notably, 24 h after transplantation, we observed a significant decrease in our patients, which was interestingly similar to the results in the control group (however, these concentrations are still higher than in the control group). What is important is that the highest FGF19 concentration in all the mentioned diseases occurs in patients before transplantation, and regardless of the primary cause of the disease, the trends of increases and decreases in this concentration 24 h after the surgery are the same. A similar decreasing tendency in FGF19 concentration after liver transplantation in HCC patients was observed by Miura et al. [13]. These scientists believe that the reason for such high concentrations of FGF19 is the presence of factors accompanying primary liver diseases, such as increased stress or chronic inflammation, eliminated after transplantation. We also consider that such a high FGF19 concentration before transplantation compared to the group of healthy subjects is associated with the critically impaired function of the failing liver. In such patients, there are disturbances in the production and circulation of bile acids. In response to this process, the intestines may increase the production of FGF19 as a compensatory mechanism [18,20,21,30,31]. After transplantation, the new, healthy liver copes much better with the metabolism of bile acids, which leads to the normalization of FGF19 concentrations. Research on FGF19 concentration before and after transplantation was also performed by the scientists Marchelek-Myśliwiec et al. [11], with the difference that, unlike us, these scientists observed changes in FGF19 concentration in patients after kidney transplantation. In this case, the study group consisted of chronic kidney disease patients (CKD) not undergoing hemodialysis, patients with CKD stage V (with hemodialysis), post-transplant patients, and healthy volunteers as a control group. In these studies, a correlation of FGF19 concentration with GFR was observed (the higher the FGF19 concentration, the lower the GFR). This research showed that FGF 19 concentrations were significantly lower after transplantation than before the surgery. Therefore, these results show that kidney transplantation (as well as liver transplantation in our studies) leads to a reduction in FGF19 levels.

Interestingly, 2 weeks after transplantation, the level of FGF19 increased 1.7-fold compared to the levels 24 h after transplantation, which may indicate that regenerative [32,33,34], proliferative, and cell survival mechanisms have been activated [35]. We observed this tendency in all disease entities. We know that FGF19 is an important factor in protecting the liver against various damaging factors (hepatoprotective function) [20,24,35,36]. We also know that FGF19 stimulates liver regeneration, promoting hepatocyte proliferation, which is crucial for liver tissue reconstruction [35,36,37,38,39,40,41]. Therefore, we believe that in the context of liver transplantation, a temporary increase in FGF19 may play an important role in improving postoperative outcomes, reducing the risk of complications, and promoting the transplant organ’s regeneration. We also wonder whether the increase in FGF19 concentration 2 weeks after transplantation compared to 24 h after this surgery is related to the fact that at the initial stage, patients are fed parenterally, and oral nutrition is started on the 2nd–3rd day. As is known, FGF19 is a factor produced by the intestines after a meal and constitutes an “impulse” for its secretion [21,42], hence the increase in FGF19 concentration in patients after 2 weeks, compared to 24 h after surgery when they are fed parenterally. An interesting aspect also seems to be the comparison of increases in FGF19 concentration in the post-transplant period. Studies by Maeda et al. [27] have already shown a trend in which HCC recurrence-free survival in patients with high levels of FGF19 is shorter than in patients with low baseline levels of this factor. This researcher considered that FGF19 may be a marker that monitors the therapeutic effect. However, this theory still requires further research and, in our case, more extended monitoring of the study group, because 2 weeks after the transplantation, we observed increases in FGF19. Still, we do not notice any statistically significant differences between our patients, regardless of the primary disease.

The available literature data found no correlation between AIH, ALD, PSC, HBV, and HCV diseases and FGF19 concentration before and after liver transplantation. We know that the overexpression of FGF19 has been observed in patients with cirrhosis, NAFLD, PBC, precancerous condition, and HCC [13,27,30] and that FGF19 is a factor whose concentration tends to increase depending on the severity of pathological conditions with the significantly higher level of FGF19 in HCC patients [27]. FGF19 increases may also correlate with basic laboratory parameters: negative correlation with albumin and Hb and positive correlation with bilirubin and AST, e.g., in patients with PBC [30]. Interestingly, we did not observe many correlations with basic laboratory parameters in our study. Nevertheless, there were several, some of which seem to be of clinical importance.

CRP and FGF19 correlations seem particularly promising. We observed three positive correlations (positive correlation of FGF19 with CRP before transplantation in AIH patients; positive correlation of FGF19 with CRP two weeks after transplantation in PBC and HCC patients). However, we also observed one negative correlation between FGF19 and CRP 24 h after transplantation in patients with HCV. We suspect that the differences in correlation between FGF19 and CRP are due to the complexity of our patients’ disease processes and their individual inflammatory responses. Studies have shown that FGF19 and CRP levels can be positively correlated in most liver diseases, but a negative correlation may occur in other conditions, such as viral infections [43,44,45]. Therefore, in our study of patients with various liver diseases, it was not surprising to observe different types of correlations or even none at all. The positive correlations likely reflect the liver’s metabolic stress regulation due to acute inflammation before transplantation [21,31] or adaptation to post-transplant hepatic stress [46]. In AIH, intense inflammation drives CRP and FGF19 production through liver damage and immunological mechanisms (pro-inflammatory cytokines stimulate FGF19 production) [47]. In other liver diseases, the pathophysiological mechanisms may differ, and the inflammatory and metabolic responses may not correlate in the same way. On the other hand, the positive correlation between CRP and FGF19 observed two weeks post-transplant in patients with PBC and HCC may be related to specific clinical and pathophysiological factors characteristic of these primary diseases. We suspect that in the case of PBC and HCC post-transplant, the new liver gradually regulates bile acid metabolism, which involves appropriate FGF19 production. However, this adaptation process is still burdened by ongoing inflammation associated with healing and immune response, leading to increased CRP. The negative correlation in HCV patients observed 24 h post-transplant is more surprising. We suspect that a strong inflammatory response to the new organ and virus suppression in HCV could reduce FGF19 production and impair bile acid metabolism shortly after transplantation [45,48,49]. Additionally, shortly after transplantation, the liver may not yet be able to fully normalize bile acid metabolism, resulting in low FGF19 levels, while CRP rises as a response to postoperative inflammation [18,50].

In addition to the above-described correlations between FGF19 and CRP, we also observed other correlations: four positive correlations (between hemoglobin and FGF19 24 h after transplantation in HCC patients, between hemoglobin and FGF19 2 weeks after transplantation in HBV patients, and between ALT/total bilirubin and FGF19 2 weeks after transplantation in the HCC group) and one negative correlation (between AST and FGF19 2 weeks after transplantation in ALD patients). A positive correlation between hemoglobin and FGF19 in HCC patients 24 h post-transplant, which we do not observe in other diseases, may be due to the unique metabolic needs and the body’s specific response to liver regeneration in the context of prior cancer [30,45,51]. It seems that the regenerative process of the liver following cancer requires much more intensive support for oxygenation, and the increase in FGF19 supports bile acid metabolism and the restoration of liver energy homeostasis [52]. In contrast, the same correlation in HBV patients two weeks post-transplant may reflect the liver’s slower adaptation due to chronic infection. In this case, the newly transplanted liver may require more time to regulate bile acid metabolism than other diseases. The rise in FGF19 is a response to this process, with higher hemoglobin levels potentially facilitating its production by ensuring better oxygenation. The positive correlation between FGF19 and ALT in HCC patients two weeks after transplantation suggests active liver cell regeneration [53]. A high bilirubin level at the same time might result from significant metabolic stress [54] or mild bile stagnation [55], which could increase FGF19 production to support the adaptation of the transplanted liver [53]. Meanwhile, in ALD patients, the negative correlation between AST and FGF19 during the same post-transplant period indicates the liver’s unique adaptation to alcohol-induced damage, where FGF19 supports regeneration without further harming liver cells [56,57].

Our research did not show statistically significant differences in the concentration of FGF19 depending on age and the period of blood collection, with one exception (a higher concentration of FGF19 before transplantation in the group under 45 years of age in PBC patients). On the one hand, the elevated FGF19 levels in this younger group may indicate a more active attempt at regeneration and compensation for disruptions in bile acid metabolism, which is critical in PBC, where chronic inflammation leads to bile duct destruction [58]. On the other hand, this process may be influenced by differences in immune system activity [59]. Younger patients may exhibit a more active immune response, leading to a more intense inflammatory process and increased FGF19 production. Thus, these differences suggest that despite the similar advancement of PBC, the younger body engages different compensatory mechanisms that may manifest as higher FGF19 levels, even when the clinical situation of the patients appears comparable.

In the course of our study, we analyzed the correlation between FGF19 levels and the duration of the surgery. The results showed no significant relationship between these parameters, except for a negative correlation observed only in patients with HCV as the primary disease. This correlation, concerning FGF19 levels 24 h after transplantation, may suggest a link to this patient group’s elevated metabolic stress characteristic.

### 3.2. Changes in FGF21 Concentration in Patients Before and After Liver Transplantation (24 h, 2 Weeks) and Comparison in Those Changes with the Healthy Control Group

Our studies show that the average FGF21 plasma concentration in the control group was lower than in patients before liver transplantation, and it was 290.59 ± 68.41 pg/mL. The concentration of FGF21 in plasma collected on EDTA from a group of healthy subjects in the study by Dostalova et al. performed using the same method and from the same manufacturer is, on average, 272.3 ± 40.00 pg/mL [25], which is similar to our result and other scientists [25,26,60,61,62,63,64,65,66,67,68,69].

Our average FGF21 concentration in the control group was lower than in patients before liver transplantation (290.59 ± 47.7 vs. 466.9 ± 408). The lower level of FGF21 in the control group is most likely due to the lack of chronic liver diseases and is associated with metabolic disorders, inflammation, and oxidative stress, which are seen in patients before liver transplantation and may cause an increase in FGF21. A healthy liver maintains FGF21 concentration at a lower level because there is no need for its increased production; there is no need to support the body’s defense mechanisms or counteract the metabolic disorders characteristic of liver diseases. [12,21,53,70,71,72,73,74,75,76,77]. Additionally, in a healthy liver, FGF21 production is tightly regulated and occurs mainly in response to specific signals, such as hunger or fat consumption. In the case of liver disease, these regulatory mechanisms may be impaired, leading to excessive FGF21 production, even without stimuli that would normally cause it [53,78,79].

Interestingly, 24 h after the transplantation, the concentration of FGF21 in our patients increased very rapidly (7-fold increase), and then gradually but slowly decreased (after 2 weeks, it was still 4.6-fold higher compared to the level before transplantation, and about 7-fold higher compared to the control group). We consider that such a rapid increase in FGF21 immediately after transplantation is influenced by both the role of FGF21 in our body and the body’s reaction to the transplant itself. FGF21 is mainly produced, and its production is regulated by many metabolic factors, such as hunger, diet, or oxidative stress [21,76,77,78]. After liver transplantation, the new organ may, therefore, begin to intensively produce FGF21 in response to the stress associated with the surgery and adaptation to the new environment in the recipient’s body [73]. Additionally, FGF21 is a factor secreted in response to tissue damage and inflammation. In this case, its increase may support repair processes and regulate protein and fat metabolism to protect the transplanted organ from further damage [73,78]. In this case, we suspect that FGF21 can be a sensitive peritransplantation parameter, probably related to the fact that it reduces the production of pro-inflammatory cytokines. Contrary to appearances, this may be a beneficial phenomenon because such rapid increases in the concentration of this parameter on the first day after transplantation may reduce the probability of damage to the newly transplanted organ in the inflammatory process that accompanies the post-transplantation period [70,80,81,82].

However, when we look at individual disease entities more closely, our studies showed that in the case of two (AIH and ALD), the concentration of FGF21 was the lowest immediately before transplantation. This value gradually increased (the highest was 2 weeks after transplantation), while in the remaining cases, the lowest concentration was before transplantation, and the highest was 24 h after its execution. Regardless of the time of blood collection and the primary disease, we found statistically significant differences in all patients. We did not find much information in the available literature data about the correlation between primary diseases and FGF21 concentration before and after liver transplantation. What we know is that the level of FGF21 in blood serum increases in patients with ALD, is not dependent on gender and age, and does not correlate with the severity of the disease assessed on the Child–Pugh and MELD scales [83]. Our studies confirm the absence of significant correlations between FGF21 levels and the MELD score across all the diseases studied, except for a positive correlation between FGF21 levels 2 weeks after transplantation and the MELD score in patients with HCV. This correlation may result from activating adaptive mechanisms in the transplanted liver in response to the increased metabolic burden and chronic inflammation caused by HCV infection. Conversely, the absence of such a correlation in other diseases may be attributed to their lesser impact on liver metabolism than HCV [83,84,85]. In the case of ALD, the systemic FGF21 concentration is also not related to liver function parameters, such as bilirubin, ALT, or AST, and with any measured adipokine (e.g., adiponectin) or inflammatory parameter, i.e., CRP. Therefore, on this basis, FGF21 is not considered a marker of liver dysfunction in these patients [83], which we can confirm because we did not find any statistically significant correlations in the period before transplantation. However, our studies showed four positive correlations (between FGF21 and CRP, 24 h after transplantation in PSC patients, between ALP/GGTP and FGF21, 2 weeks after transplantation in ALD patients, and between CRP and FGF21, 2 weeks after transplantation in PSC patients) and three negative correlations (between FGF21 and ALT/hemoglobin, 24 h after transplantation in HCV patients, between albumin and FGF21, 2 weeks after transplantation in PSC patients).

We consider that the differences in the relationships between FGF21 and parameters such as CRP, Hb, albumin, ALT, ALP, and GGTP result from the complex interplay of disease pathophysiology, inflammation, liver function, and individual patient variability. The negative correlation between FGF21 and ALT and Hb levels in HCV patients 24 h post-transplant may suggest that as FGF19 supports hepatocyte regeneration [86,87,88] and improves bile acid metabolism [89,90], ALT and hemoglobin levels decrease. This reflects an adaptation to the new post-transplant state, with a mechanism that may differ from other diseases due to previous virus-induced damage. We observed a positive correlation between FGF21 and CRP in PCC patients during this same period. In this case, we believe the combined effect of an intense inflammatory response (characteristic of autoimmune diseases [91,92]) and the activation of metabolic pathways related to FGF21 is responsible. The positive correlation between ALP, GGTP, and FGF21 observed two weeks post-transplant in ALD patients may be the body’s response to metabolic stress affecting the liver at this time [93,94]. In response to transplant-related stress, the body may increase FGF21 production as a protective factor, while the rise in these enzymes may reflect the inflammatory response that the patient experiences after the transplant [95]. Similarly, the positive correlation between CRP and FGF21 and the negative correlation between albumin and FGF21 in PSC patients two weeks after liver transplantation results from the interaction between chronic inflammation, impaired liver function, and the body’s response to the transplant. These unique correlations, which may not occur in other liver diseases, are likely related to the specific pathophysiology of PSC, including the mechanisms of cholestasis and chronic inflammation [96].

Studies on the concentration of FGF21 were also carried out in patients infected with HBV [97]. Interestingly, serum FGF21 levels and biochemical markers of liver injury were significantly higher in acute hepatitis B (AHB) infection and significantly lower in chronic hepatitis B (CHB) patients in comparison to the control group. Interestingly, a drastic increase in FGF21 concentration in patients who developed HCC due to CHB was observed. From the point of view of our research, we also observed a difference in FGF21 concentration before transplantation between HCC and HBV patients (*p* = 0.0017), which means that regardless of whether HCC developed as a result of CHB or with other causes, FGF21 concentrations are higher in HCC patients, compared to CHB. Similar studies have also been conducted in patients with HCV. The group of these investigators showed that serum FGF21 levels were significantly higher in patients with chronic HCV compared to the control group [98]. In the case of these studies, no statistically significant differences in FGF21 concentration were observed depending on gender and between patients with the inflammatory conditions of the different activity and degrees of fibrosis. In our research, in the case of HCV, we noticed only one statistically significant difference depending on gender (24 h after transplantation, *p* = 0.014). Still, considering that at other time points or in any other primary disease, we do not observe differences in FGF21 plasma concentration according to gender, then we assume such differences do not exist.

Our research did not also show statistically significant differences in the concentration of FGF21 depending on age and the period of blood collection, with two exceptions (a higher concentration of FGF21 before transplantation in the group under 45 years of age in PBC patients; a lower concentration of FGF21 before transplantation in the group under 45 years of age in AIH patients). Higher plasma FGF21 levels in younger PBC patients compared to older ones before transplantation, alongside the absence of similar differences in other liver diseases, suggest unique characteristics specific to this condition. We suspect that the body may attempt to compensate for liver damage in younger PBC patients by increasing FGF21 production. The body’s response might not be as intense in diseases that do not lead to such specific bile duct dysfunctions [90]. Compared to older individuals, younger patients may also have a stronger immune response to chronic bile duct inflammation [99], potentially leading to higher FGF21 production as well. Interestingly, we did not observe such relationships either 24 h post-transplant or two weeks later, so we have no grounds to conclude that age affects the effectiveness of the transplant or the return to metabolic normalcy. We suspect that in younger individuals, this mechanism may not be directly associated with FGF21 production, or the metabolic response to liver damage may be less intense. In older patients, however, advanced liver changes might lead to higher FGF21 production as a response to damage like in another liver disease [78,97,100,101,102].

Interesting conclusions, especially in the context of our research, were reached by the team of scientists Ye et al. [12]. Their study showed that the serum levels of FGF21 in patients after transplantation, regardless of the primary disease compared to healthy subjects, are about a 20–25-fold increase already 2 h after reperfusion and gradually returned after about 24 h to the baseline level. These data suggest that significant increases in serum FGF21 occur long before massive hepatocyte damage induced by IRI after transplantation. Our research (24 h after transplantation) shows a significant difference in FGF21 concentration compared to the immediate period before transplantation. We wonder whether the FGF21 levels we observed 24 h after transplantation can already be associated with a protective function after IRI, which most likely occurred due to the transplantation.

These studies [12,83,97,98], as well as our observations, suggest that the increase in FGF21 concentration before transplantation in our patients compared to healthy subjects is closely related to active liver injury and the impairment of its synthetic function. In contrast, the increase in the FGF21 level 24 h after transplantation and remaining at a lower level 2 weeks after surgery indicates an adaptive/protective response in the transplanted liver. In the case of AIH and ALD, a significant increase in FGF21 2 weeks after transplantation differs from previously presented observations. However, further research is needed to explain the cause-and-effect relationship that has been observed.

### 3.3. Conclusions

Our study revealed dynamic changes in FGF19 and FGF21 levels before and after liver transplantation, highlighting their critical and diverse roles in the adaptation and regeneration processes of the transplanted organ. The decrease in FGF19 levels immediately after transplantation suggests the resolution of prior liver dysfunction, while its subsequent gradual increase indicates potential support for regenerative processes. Similarly, the rapid increase in FGF21 levels shortly after the procedure emphasizes its role in the metabolic stress response and adaptation, with sustained elevated levels reflecting active repair and adaptive processes. Differences in correlations with basic biochemical parameters depending on the primary liver disease suggest unique functions of FGF19 and FGF21 in various clinical contexts. Positive correlations of FGF19 with CRP in patients with AIH, PBC, and HCC at different post-transplant stages indicate its association with inflammatory processes and metabolic regulation. Conversely, the negative correlation of FGF19 with CRP in HCV patients reflects specific inflammatory mechanisms resulting from viral damage. For FGF21, correlations with CRP, ALT, and albumin at different times post-transplant confirm its role in tissue damage response, regeneration, and metabolic adaptation. These observations could serve as valuable tools in assessing the risk of post-transplant complications, such as graft rejection, failure, or regeneration. A deeper understanding of the functions and processes regulated by FGF19 and FGF21 may contribute to more effective diagnostics, the optimization of post-transplant care, and the development of targeted therapeutic approaches.

### 3.4. Limitations

1. This study includes a relatively small number of patients, which may limit the statistical power and generalizability of the findings.

2. This study was conducted at a single medical institution, which may limit its applicability to broader populations in different geographic regions or healthcare settings.

## 4. Materials and Methods

### 4.1. Study Group

The study group included 84 patients (35 women and 49 men) aged 20 to 68 who underwent liver transplantation and in whom no complications after this surgery were observed. All patients were discharged home 2–3 weeks after the surgery. In addition to age and the lack of complications after transplantation, the inclusion criterion for the study was also the specified immunosuppressive regimen (in anhepatic phase: methylprednisolone 500 mg iv + basiliximab 20 mg iv (1st dose), after 12 h: methylprednisolone 250 mg iv, 1st POD: methylprednisolone 125 mg iv, from 2nd POD: Prednisone 20 mg po, on 4th POD: basiliximab 20 mg iv (2nd dose), and from 5th POD: tacrolimus 2 × 2 mg (dosage dependent on tacrolimus serum level, desired level 7–10 mg/mL). Transplantations were performed at the Department and Clinic of General, Transplant, and Liver Surgery of the Medical University of Warsaw. Demographic and laboratory data on analyzed patients are summarized in Table 3. This study was approved by The Bioethical Commission at the Warsaw Medical University (No. KB/224/2016). The study group was divided according to disease entities: AIH (n = 13), ALD (n = 13), PBC (n = 9), PSC (n = 9), HBV (n = 12), HCV (n = 10), and HCC (n = 19). The control group consisted of 40 (22 men and 18 women) healthy individuals aged 27 to 72 years. Material from healthy individuals was collected at the Pomeranian Medical University in Szczecin based on the consent of the bioethics committee No. KB-0012/10/17.

### 4.2. Study Material

For the tests, peripheral blood was collected in the antecubital vein three times (in tubes with a K_2_EDTA to obtain a plasma fraction and in tubes with a clotting activator to obtain a serum fraction). The material was collected immediately before liver transplant surgery, on the first day after surgery, and two weeks after transplantation. Peripheral blood counts in each patient’s samples collected on K_2_EDTA were performed. Then, the blood collected on K_2_EDTA and serum were centrifuged. K_2_EDTA plasma was transferred to new tubes and kept at −80 °C until the samples were transported on dry ice to the Medical Analytics Department of the Pomeranian Medical University, where FGF19 and FGF21 analyses were performed. Serum was used to perform several basic biochemicals (CRP, total bilirubin, ALP, ALT, AST, GGTP, and albumin). Peripheral blood from healthy volunteers was collected in the antecubital vein in the morning and fasting (in tubes with a K_2_EDTA to obtain a plasma fraction). Volunteers had to meet the eligibility criteria, such as no chronic diseases and negative test results for infectious diseases (HIV, HBV, HCV). The obtained plasma was transferred to new tubes and stored at −80 °C until assayed.

### 4.3. Assay Procedure

Reagents and examined plasma were equilibrated to room temperature before analysis. Plasma FGF19 and FGF21 concentrations were measured in all collected samples using the enzyme-linked immunosorbent assay reagent kits (Human FGF19 and Human FGF21 sandwich ELISA kits from BioVendor). All procedures were conducted in duplicate according to the manufacturer’s instructions. Reaction products were measured using an EnVision microplate reader (Perkin Elmer, Waltham, MA, USA) at 450 nm, and their concentration was calculated based on standard curves created based on the standard solutions of specific concentrations included in the kit.

Most laboratory parameters such as total bilirubin, alkaline phosphatase (ALP), alanine aminotransferase (ALT), aspartate aminotransferase (AST), gamma-glutamyl transpeptidase (GGTP), and albumin were determined in serum using the spectrophotometric method and the Cobas 6000 module c501 biochemistry analyzer from Roche (Roche, Basel, Switzerland). C-reactive protein (CRP) was determined by immunoturbidimetric assay in serum on the Cobas 6000 module c501 biochemistry analyzer from Roche (Roche, Basel, Switzerland). The optical detection method in K_2_EDTA whole blood determined hemoglobin (Hb) on the MEK-7300 hematology analyzer (Nihon Kohden, Tokyo, Japan). All determinations were performed according to the manufacturer’s instructions at the same time points (before transplantation (0 h), 24 h after transplantation, and 2 weeks after transplantation) in the Central Laboratory of the University Clinical Center of the Medical University of Warsaw.

### 4.4. Statistical Analysis

Results were evaluated using the Statistica PL 13 statistical package (StatSoft, version 13.0, Kraków, Poland). The Shapiro–Wilk test was performed to assess the distribution of variables. Because the compared factors were time-related variables, their differences were analyzed based on paired tests. The Friedman ANOVA and Kendall compliance tests determined possible changes in the concentration of all tested factors. The Kruskal–Wallis test assessed the concentration change between the control group and the tested time points. Depending on the normal distribution, the T-Test or Wilcoxon signed-rank test evaluated the differences in parameter concentrations in studied time points. The T-test for independent groups or the Mann–Whitney U test was performed depending on the normal distribution to check the statistical significance between the control group and the studied time points. The Mann–Whitney test was made to examine the significance of differences in the concentration of FGF19 and FGF21 between men, women, and the control group and assess the concentration of the tested factors depending on the age group (under and equal to 45 years and over 45 years), as well as the control group. The Kruskal–Wallis test was performed to assess the differences in the concentrations of FGF19 and FGF21 in the examined periods between the studied diseases. Additionally, the correlations of basic laboratory parameters (CRP, Hb, total bilirubin, ALP, ALT, AST, GGTP, albumin) with the tested concentration of FGF19 and FGF21 were performed using the Pearson or Spearman’s correlation test depending on the normal distribution, in parameter concentrations. Moreover, Spearman’s correlation test assessed the correlation between FGF19 and FGF21 levels at all time points with the MELD score and length of surgery time.

## Figures and Tables

**Figure 1 ijms-26-01299-f001:**
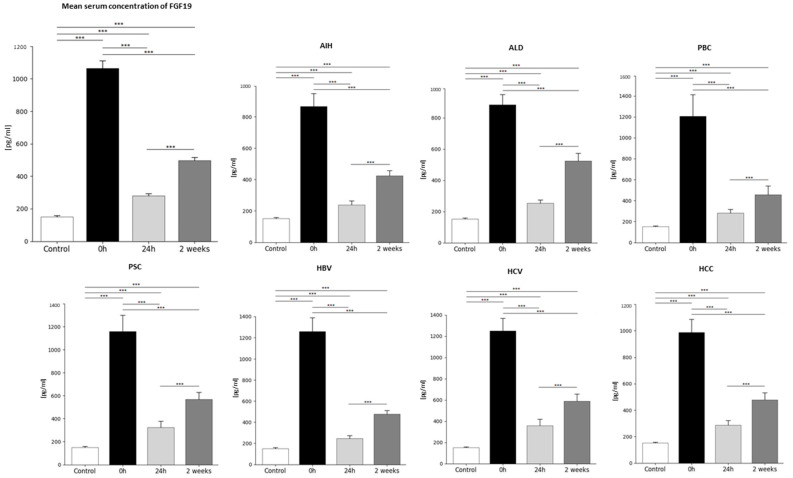
The concentration of FGF19 in plasma in patients in the period before (0 h) and after the transplantation (24 h, 2 weeks) and in the control group [pg/mL]. Plasma concentration of FGF19 was the highest at 0 h and decreased 3.8-fold 24 h after transplantation, and then a 1.7-fold increase 2 weeks after transplantation. The lowest FGF19 concentrations were observed in the control group and were 7-fold lower compared to the FGF19 levels in patients before transplantation, 1.9-fold lower compared to the levels 24 h after transplantation, and 3.3-fold lower than 2 weeks after transplantation. FGF19 concentration was evaluated using an ELISA assay. In all examined diseases, as in the aggregate results, in patients qualified for transplantation, the highest concentration of FGF19 occurred before transplantation (0 h) and the lowest 24 h after transplantation; however, the lowest concentration of FGF19 levels was observed in the control group. Data were compared with the Friedman ANOVA test and the Kendall, as well as the Kruskal–Wallis test. The T-Test or Wilcoxon signed-rank test assessed the differences in parameter concentrations, depending on the normal distribution. The T-test for independent groups or the Mann–Whitney U test was performed depending on the normal distribution to check the statistical significance between the control group and the studied time points. *p*-values below 0.05 were considered statistically significant. Bars indicate the mean ±  standard deviation (SD), *** *p* <0.0001. The group sizes were as follows: AIH (n = 13), ALD (n = 13), PBC (n = 9), PSC (n = 9), HBV (n = 12), HCV (n = 10), and HCC (n = 19).

**Figure 2 ijms-26-01299-f002:**
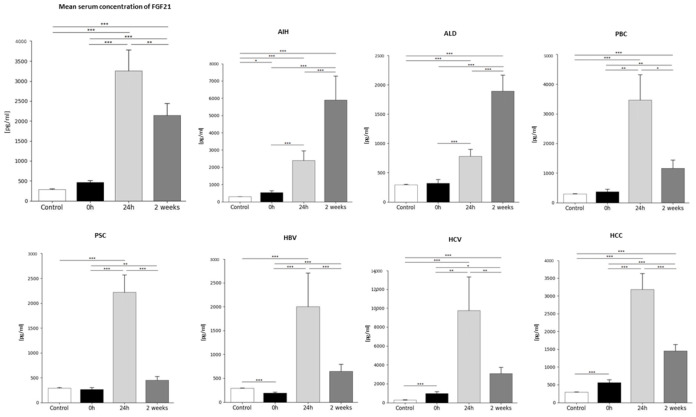
The concentration of FGF21 in plasma in patients in the period before (0 h) and after the transplantation (24 h, 2 weeks) and in the control group [pg/mL]. Plasma concentration of FGF21 was the lowest at 0 h and increased 7-fold 24 h after transplantation and then 0.7-fold, decreasing 2 weeks after transplantation. In all examined diseases, the lowest concentration of FGF21 occurred before transplantation (0 h), and the highest occurred 24 h after transplantation in the case of PBC, PSC, HBV, HCV, and HCC, and 2 weeks after transplantation in the case of AIH and ALD. The lowest FGF21 concentrations were observed in the control group and were 1.6-fold lower compared to the FGF19 levels in patients before transplantation, 11.2-fold lower compared to the levels 24 h after transplantation, and 7.4-fold lower than 2 weeks after transplantation. FGF21 concentration was evaluated using an ELISA assay. Data were compared with the Friedman ANOVA test and the Kendall, as well as the Kruskal–Wallis test. The T-Test or Wilcoxon signed-rank test assessed the differences in parameter concentrations, depending on the normal distribution. The T-test for independent groups or the Mann–Whitney U test was performed depending on the normal distribution to check the statistical significance between the control group and the studied time points. *p*-values below 0.05 were considered statistically significant. Bars indicate the mean  ±  standard deviation (SD), * *p* < 0.05, ** *p* < 0.01, and *** *p* < 0.0001. The group sizes were as follows: AIH (n = 13), ALD (n = 13), PBC (n = 9), PSC (n = 9), HBV (n = 12), HCV (n = 10), and HCC (n = 19).

**Figure 3 ijms-26-01299-f003:**
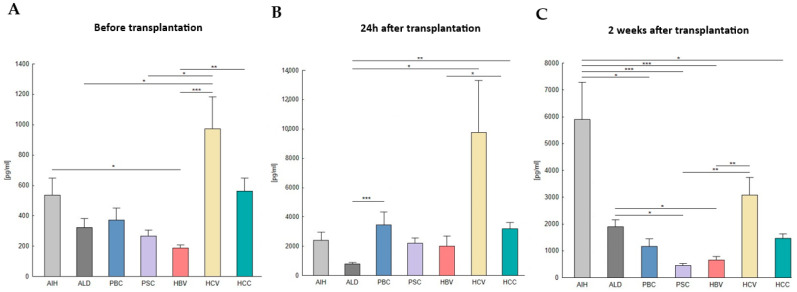
FGF21 concentration in the plasma of patients before (0 h) and after liver transplantation (24 h, 2 weeks), depending on the disease. (**A**) Before transplantation (0 h), the highest plasma concentration was observed in HCV patients, and the lowest was observed in the HBV group; (**B**) 24 h after transplantation, the highest plasma concentration was observed in HCV patients, and the lowest was observed in the ALD group; (**C**) 2 weeks after transplantation, the highest plasma concentration was observed in AIH patients, and the lowest was observed in the PSC group. FGF21 concentration was evaluated using an ELISA assay. The Kruskal–Wallis test was performed to assess the differences in FGF21 concentrations. *p*-values below 0.05 were considered statistically significant. Bars indicate the mean  ±  standard error of the mean (SEM), * *p* < 0.05, ** *p* < 0.01, and *** *p* < 0.0001.

**Figure 4 ijms-26-01299-f004:**
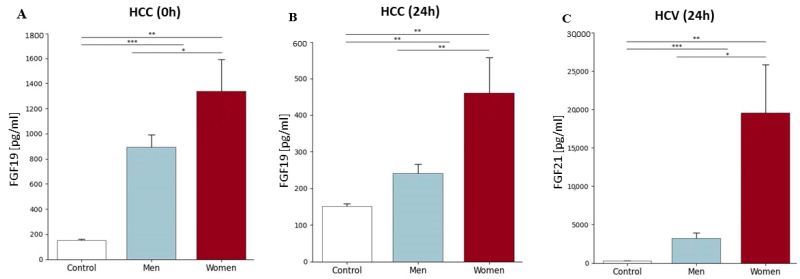
(**A**,**B**) FGF19 concentration in plasma of HCC patients depending on gender before transplantation (0 h) and 24 h after transplantation and in the control group. In both periods, higher FGF19 levels are observed in women. (**C**) FGF21 concentration in HCV plasma, depending on gender, 24 h after transplantation. FGF19 and FGF21 concentrations were evaluated using an ELISA assay. The Mann–Whitney test was performed to assess the differences in parameter concentrations. *p*-values below 0.05 were considered statistically significant. Bars indicate the mean  ±  standard deviation (SD), * *p* < 0.05, ** *p* < 0.01, and *** *p* < 0.0001. In the HCC group, the number of men was 15, and women, 4. However, in the case of the HCV group, the number of men was 5, and women, 5.

**Figure 5 ijms-26-01299-f005:**
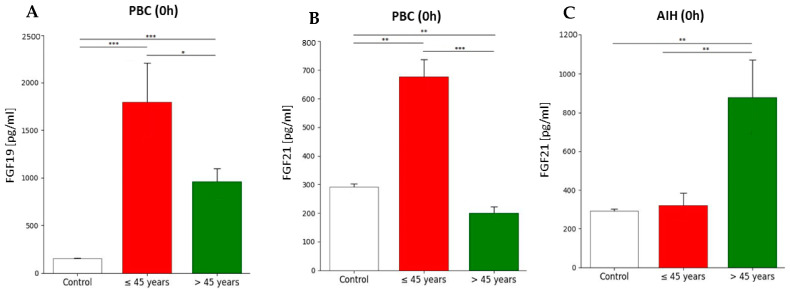
Plasma concentration of FGF19 and FGF21 in patients and control group depending on age. (**A**) Plasma concentration of FGF19 in PBC patients was higher in younger patients before transplantation. (**B**) Before transplantation, plasma concentration of FGF21 in PBC patients was higher in younger patients. In the AIH group (**C**), plasma concentration of FGF21 was higher in patients over 45 years of age. FGF19 and FGF21 concentration were evaluated using an ELISA assay. The T-test or Mann–Whitney U test was performed depending on the normal distribution to assess the differences in parameter concentrations. *p*-values below 0.05 were considered statistically significant. Bars indicate the mean ± standard deviation (SD), * *p* < 0.05, ** *p* < 0.01, and *** *p* < 0.0001. The number in the group of people ≤ 45 was 23, while in the group over 45, it was 59; in PBC (n = 3 for ≤45 groups, n = 5 for >45 groups), and in AIH (n = 8 for ≤45 group, n = 5 for > 45 groups).

**Figure 6 ijms-26-01299-f006:**
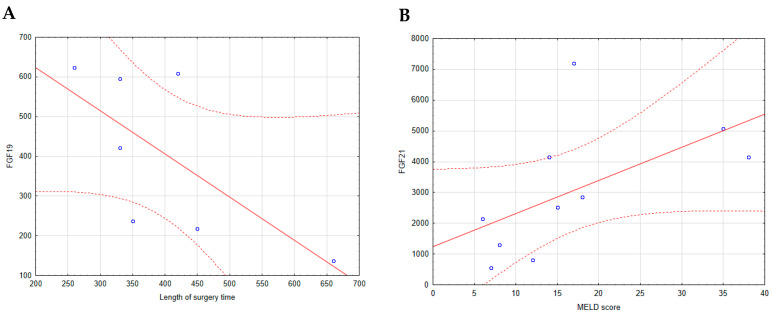
FGF19 and FGF21 levels correlate with the MELD score and length of surgery time in HCV patients. (**A**) Correlation between levels of FGF19 24 h after transplantation and length of surgery time in HCV patients. (**B**) Correlation between levels of FGF21 2 weeks after transplantation and MELD score of HCV patients. The analysis was performed using the Spearman correlation test. The number in the HCV group of people was 10. The solid red line is the regression line representing the trend of the relationship between the variables. The dashed red line is the confidence interval line, showing the range of uncertainty around the regression line.

**Table 1 ijms-26-01299-t001:** FGF19 plasma concentration (mean ± SD) in individual diseases was the direct cause of transplantation before (0 h) and after transplantation (24 h, 2 weeks) and in the control group [pg/mL].

	FGF19 [pg/mL]				
Disease	Time of Blood Collection				*p* Value
	Control Group	0 h	24 h	2 Weeks	
AIH	151.0 ± 47.7	865.9 ± 286	237.8 ± 95	423.8 ± 118	<0.0001
ALD		884.9 ± 214	252.6 ± 75	523.9 ± 162	<0.0001
PBC		1206.8 ± 580	280.9 ± 104	457.7 ± 235	<0.0001
PSC		1160.1 ± 396	325.4 ± 154	569.8 ± 173	<0.0001
HBV		1259.3 ± 430	245.3 ± 94	476.7 ± 120	<0.0001
HCV		1250 ± 352	356.5 ± 182	589.9 ± 201	<0.0001
HCC		986.5 ± 425	287.8 ± 142	479.4 ± 222	<0.0001

**Table 2 ijms-26-01299-t002:** FGF21 plasma concentration (mean ± SD) in individual diseases was the direct cause of transplantation before (0 h) and after transplantation (24 h, 2 weeks) and in the control group [pg/mL].

	FGF21 [pg/mL]				
Disease	Time of Blood Collection				*p* Value
	Control Group	0 h	24 h	2 Weeks	
AIH	290.59 ± 47.7	534.7 ± 387	2389.1 ± 1983	5897 ± 4808	<0.0001
ALD		321.4 ± 205	781.8 ± 401	1892.2 ± 915	<0.0001
PBC		371.9 ± 226	3464.1 ± 2435	1165 ± 798	<0.0001
PSC		266 ± 108	2220.2 ± 992	449.1 ± 221	<0.0001
HBV		189 ± 65	2004.5 ± 2326	649.9 ± 485	<0.0001
HCV		973 ± 631	9758.3 ± 10,635	3073.4 ± 1979	<0.0001
HCC		561.7 ± 373	3182.6 ± 1960	1458.9 ± 762	<0.0001

**Table 3 ijms-26-01299-t003:** Demographic and laboratory features of analyzed patients.

	Time of Blood Collection		
Parameters	0 h	24 h	2 Weeks
Gender (M/F)	35/49	35/49	35/49
Age (mean ± SD)	50.9 ± 13	50.9 ± 13	50.9 ± 13
CRP (mean ± SD NR < 5 mg/L)	11.1 ± 12.3	41.7 ± 38.9	17.2 ± 19.6
Hb (mean ± SD, NR 12–16 g/dL)	11.9 ± 2.2	10.4 ± 1.25	10.2 ± 1.21
Total bilirubin (mean ± SD, NR 0.2–1.1 mg/dL)	6.2 ± 9.4	3.2 ± 3.3	1.9 ± 3.35
ALP (mean ± SD, NR 40–120 IU/L)	165 ± 125	272 ± 158	248 ± 216
ALT (mean ± SD, NR 5–40 IU/L)	91 ± 117.4	293 ± 226	123 ± 114
AST (mean ± SD, NR 5–40 IU/L)	105 ± 123	115 ± 161	47 ± 56
GGTP (mean ± SD, NR 5–40 IU/L)	152 ± 274	781 ± 520	567 ± 500
Albumin (mean ± SD, NR 3.8–4.2 g/dL)	3.6 ± 0.64	3.1 ± 0.42	3.3 ± 0.54

Abbreviations: 0 h—before transplantation; 24 h—24 h after transplantation; 2 Weeks—2 weeks after transplantation; M—men; F—women; CRP—C-reactive protein; Hb—hemoglobin; ALP—alkaline phosphatase; ALT—alanine aminotransferase; AST—aspartate aminotransferase; GGTP—gamma-glutamyl transpeptidase; SD—standard deviation; NR—normal range.

## Data Availability

The data presented in this study are available on request from the corresponding author. The data are not publicly available due to institutional privacy restrictions.

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
