# Peer review of "The Impact of Human Liver Transplantation on the Concentration of Fibroblast Growth Factors: FGF19 and FGF21"

_ijms, 2025, doi:10.3390/ijms26031299_

Round 1
Reviewer 1 Report (Previous Reviewer 2)
Comments and Suggestions for Authors
I have read the changes made to the manuscript according to my and others suggestions. I now find this manuscript suitable for publication. Below my specific concerns from first round, authors replies and my conclusion to those.
Point 1: Some of the sentences are rather long and, to my mind, discussion section could benefit from tightening.
Response 1: To comply with this Reviewer suggestion, we tried to shorten the sentences and tighten discussion section. However, we would like to point out that the discussion section has undergone a fairly large transformation due to the fact that the control group was introduced to our research. We shortened the previous version of the discussion to a significant extent (about 40%), but the fact that the control group was introduced to our research involved taking up new threads, comparisons and conclusions that we consider necessary at this stage.
OK. I still find discussion rather long, but as explained it is now long partly due to introduction of control groups. This makes sense. This time I find sentences somehow easier to read.
Point 2: Results are mainly explanatory and some of the conclusions/assumptions are bold. For instance: we assume that they may equalize later when the physiological homeostasis of the transplanted organ is restored (reduction of inflammation). This could be easily measured. However, I agree with authors on this: Therefore, a better under-standing of the functions and processes affected by these factors may contribute to more effective diagnosis of patients and the selection of more effective treatment methods
Response 2: We realize that some of our conclusions/assumptions are bold and we know that to support our assumptions we could examine these parameters in our patients in the future.. However, it would be very difficult for a few reasons:
1. This study was approved by The Bioethical Commission at the Warsaw Medical University and was limited only to monitoring patients at the Transplantation Centre, not outside it.
2. Patients qualified for transplantation came from all over Poland and at the time of leaving the main transplantional centre after about 2 weeks were already monitored in transplant clinics in their cities, or cities located closest to their place of residence, often hundreds of kilometers away from the transplant site. Therefore, taking blood from them and transporting it to us would be very expensive and currently exceeds our financial possibilities.
I understand these reasons. Manusricpt is now better.
Based on the suggestions of another Reviewer, we introduced a control group to our study. This allowed us to somewhat better understand and explain the trends of increases and decreases in FGF19 and 21 levels at individual time points in patients and to compare them at each stage with the control group. We have introduced several new comments in the discussion section and we hope that these data will enrich the general understanding of these mechanisms (Please see: page 8, lines 319-321, 324-337, 341-348, page 9, lines 363-376,
405-406, page 10, lines 407-423, 427-437).
That was a good suggestion and manuscript is now better.
Point 3: Statistical tests are mentioned in results section. This could be only in method section.
Response 3: Thank you for your attention. We have moved the descriptions of statistical tests to the methods section only, to maintain consistency and clarity in the presentation of results. Thank you for your valuable advice (Please see: 4.4. Statistical Analysis - page 14, lines 600-610
OK now
Point 4: Why results are expressed as mean +-SEM. Why not SD?
Response 4: Thank you for your valuable suggestion regarding the presentation of results in graphs. We have followed your advice and decided to express the results as mean ± SD, instead of mean ± SEM. We agree that the SD better reflects the dispersion of the data and the variability of the measurements, which gives a clearer picture of the variability in the study group. All corrected standard deviations in each section are in yellow.
OK now
Point 5: Why table 1 and 2 and figure 1 and 2 have same results? Figures may be enough.
Response 5: Thanks for pointing that out, it is indeed the same data and figures would most likely be sufficient, but we believe that each of these forms has its own unique advantages that allow for a clearer and more effective communication of information. Tables provide a precise presentation of numerical research results for the more discerning, while figures help identify and visualize general trends depending on the data. We also believe that, depending on the audience, some people better absorb information in a visual form, while others in a numerical form. We believe that presenting results in both forms allows for reaching a wider audience, increasing the effectiveness of scientific communication.
I accept if ok for editor
Point 6: 84 patients: What were the inclusion criteria? Selection method?
Response 6: 84 is the number of patients we managed to qualify for the study within a specified time, assuming specific criteria for us (we had a specified time from when to when we could obtain biological material from patients). These were age (>18 and <70) and the lack of complications after transplantation, including graft rejection, failure of the graft to take action or failure of the transplanted organ, hepatic artery thrombosis, portal vein thrombosis, hepatic vein stenosis, etc.
We also wanted the same immunosuppression regimen (most commonly used in our patients in this situation):
• Anhepatic phase: methylprednisolone 500 mg iv + basiliximab 20 mg iv (1st dose)
• After 12hours: methylprednisolone 250 mg iv
• 1st POD: methylprednisolone 125 mg iv
• From 2nd POD: Prednisone 20 mg po
• On 4th POD: basiliximab 20 mg iv (2nd dose)
• From 5th POD: tacrolimus 2x2 mg (dosage dependent on tacrolimus serum level, desired level 7-10 mg/ml)
We have added important information about inclusion criteria in the Materials and methods section (Please see: page 12, lines 535-542).
OK now
Point 7: I found two typos: trans-plantation in table 1 heading and nubmer of men… Fig 3 subheading
Response 6: We would like to thank you for drawing attention to these typos, which we have corrected of course in green Please see:
• page 3, lines 109 - trans-plantation in table 1 heading (corrected, now is transplantation)
• page 6, line 248 - nubmer of men… Fig 3 subheading (corrected, now is number)
We would also like to mention that the paper has been checked in terms of the correctness of the English language by a person with a B2 level certificate. The language corrections are marked in green. If further corrections are needed, please let me know
OK now
Author Response
Please familiarize yourself with the attachment.

Reviewer 2 Report (New Reviewer)
Comments and Suggestions for Authors
While I find the manuscript well-written, I find that there is not enough new information to support the length of the paper.
Your premise of studying FGF 19 and 21 started out promising, but other than showing that they increased or decreased post-transplant or varied by etiology, there was not much else.
1) When you looked at FGF 19 and 21 (Tables 1 & 2) levels pre- and post-transplant, there was nothing that was not unexpected here except the direction of the changes. I would have expected the significant p-values across the board - you went from a very sick liver (assumption since a transplant was apparently required) to a non-sick liver (assume viable donor liver).
2) Why did they change? Yes, the subject was transplanted. You could have looked at ischemia time, length of illness prior to transplant, the amount of change, the amount of change per etiology, length of surgery time, length of hospital stay, need for CVVH post-TX, etc). This is the major piece missing from your paper as basing you results against a normal population, with so many different etiologies and so many different unknown variables, you can only speculate.
3) You compared the results against normal controls, yet none of your subjects were normal. I would suggest that your "normal subjects" should have been subjects who had earlier stage disease to determine how the disease affects the levels of the FGF 19 and 21 prior to the need for transplant.
4) You did not provide grade of severity of illness of any of your subjects (MELD, Kings, fibrosis, cirrhosis, etc.). These parameters could possibly give some insight into your pre-transplant FGF 19 and 21 values.
5) Did you consider looking at a longer post-transplant time point to see when /if the FGF 19 and 21 values would return to a "normal" level?
6) Were there any complications with any of the subjects post-transplant that might have affected the results? See #2.
7) Did you consider looking at the amount of change and compare those by etiology for pre- and post-transplant and transplant over time.
8) You looked at several different etiologies and this is fine. However, they all act differently. Therefore, as I mentioned earlier, you should have controls based on subjects who have the disease in its earlier stages (ie prior to needing a transplant). Example: HBV/HCV subjects who are known chronic subjects that have early fibrosis, HCC subjects who are pre-surgery for a tumor removal; ALD subjects with known Hx and known fibrosis, etc. These subjects then need to be compared to the normal subjects and then to your pre-transplant values.
9) Just a request, please be consistent with your numbers. I found it quite distracting when some contained commas (9302,1) and others contained periods (9302.1).
Author Response
Please familiarize yourself with the attachment.

Reviewer 3 Report (New Reviewer)
Comments and Suggestions for Authors
Thank you for giving me the opportunity to review the manuscript titled "The impact of Human Liver Transplantation on the Concentration of Fibroblast Growth Factors: FGF19 and FGF21".
This is a very interesting topic. but there are several concerns.
1. The positive correlation between FGF19 and CRP seems to have clinical significance. However, there were instances where a negative correlation was observed as well. Given that CRP is known to be associated with patient prognosis in previous studies, an explanation of why the relationship between FGF19 and CRP varies depending on the situation would be important.
2. An explanation is needed for why the relationships between FGF19 and parameters like CRP, Hb, ALT, and total bilirubin differ across disease groups. These inconsistent relationships make it difficult to interpret the results and apply them clinically.
3. The correlations between FGF21 and CRP, ALP, and GGTP seem to have clinical significance. However, an explanation is needed for why the relationships between FGF21 and these parameters differ across disease groups. These inconsistent relationships make result interpretation and clinical application challenging.
4. Why were the correlations between FGF19, FGF21, and basic laboratory parameters not analyzed for the entire patient group?
5. An explanation for the relationship between age and FGFs would be helpful.
6. The expressions "3,8 fold" in line 91, "1,7 fold" in line 127, "1,9 fold" in line 252, and "2,7 fold" in line 260 appear to be typos. It would be more appropriate to use a comma in these cases.
Author Response
Please familiarize yourself with the attachment.

Round 2
Reviewer 2 Report (New Reviewer)
Comments and Suggestions for Authors
I while I still feel that there areas in which this study could be improved, the explanations provided were reasonable at this stage. I hope to see further studies in the future.
Author Response
Please familiarize yourself with the attachment.

Reviewer 3 Report (New Reviewer)
Comments and Suggestions for Authors
Thank you for your well-considered and detailed responses to comments. I appreciate the thoroughness with which you've addressed each point, and your explanations have clarified several aspects of the study. To strengthen the manuscript and enhance its credibility, I recommend incorporating the following study limitations.
1. The study includes a relatively small number of patients, which may limit the statistical power and generalizability of the findings.
2. The study was conducted at a single medical institution, which may limit its applicability to broader populations in different geographic regions or healthcare settings.
Author Response
Please familiarize yourself with the attachment.

This manuscript is a resubmission of an earlier submission. The following is a list of the peer review reports and author responses from that submission.
Round 1
Reviewer 1 Report
Comments and Suggestions for Authors
Hi, I am attaching a marked up version of your manuscript where yellow high lights indicate where alternate wording is suggested, or where there are obvious spelling mistakes. I have embedded comments in red text for visibility. The comments related to the content adjacent to them.
I have not made comments to the Discussion section, but it is over long. The design of the comparisons made in this paper do not allow for a clear assessment on what the changes in levels of FGF19 and FGF21 mean in the context of a healing transplanted liver. Additional factors that are important to include are the immunosuppression regime supporting the transplants, i.e. how different are they from each other across the patients. Additional insights could be gathered if serum measure results were published and plotted rather that regression values quoted.

As mentioned above, the marked up file highlights areas in the text where the english expressions could be altered for clarity or to be more succinct.
Author Response
Detail response in attachment

Reviewer 2 Report
Comments and Suggestions for Authors
Authors have performed very straightforward analysis on the concentration of Fibroblast Growth Factors: FGF19 and FGF21 before and soon after liver transplantation. Methodologically research is clear, and manuscript is well written. Some of the sentences are rather long and, to my mind, discussion section could benefit from tightening. Results are mainly explanatory and some of the conclusions/assumptions are bold. For instance: we assume that they may equalize later when the physiological homeostasis of the transplanted organ is restored (reduction of inflammation). This could be easily measured. However, I agree with authors on this: Therefore, a better under-standing of the functions and processes affected by these factors may contribute to more effective diagnosis of patients and the selection of more effective treatment methods.
I have only few concerns.
1. Statistical tests are mentioned in results section. This could be only in method section.
2. Why results are expressed as mean +-SEM. Why not SD?
3. Why table 1 and 2 and figure 1 and 2 have same results? Figures may be enough.
4. 84 patients: What were the inclusion criteria? Selection method?
I found two typos:
· trans-plantation in table 1 heading
· nubmer of men… Fig 3 subheading
Author Response
Detail response in attachment
